# Color Stability Assessment of Single- and Multi-Shade Composites Following Immersion in Staining Food Substances

**DOI:** 10.3390/dj12090285

**Published:** 2024-09-04

**Authors:** Vittorio Checchi, Eleonora Forabosco, Giulia Della Casa, Shaniko Kaleci, Luca Giannetti, Luigi Generali, Pierantonio Bellini

**Affiliations:** 1Department of Surgery, Medicine, Dentistry and Morphological Sciences with Transplant Surgery, Oncology and Regenerative Medicine Relevance, University of Modena & Reggio Emilia, 41100 Modena, Italy; eleonora.forabosco@unimore.it (E.F.); 304222@studenti.unimore.it (G.D.C.); shaniko.kaleci@unimore.it (S.K.); luca.giannetti@unimore.it (L.G.); luigi.generali@unimore.it (L.G.); pierantonio.bellini@unimore.it (P.B.); 2Clinical and Experimental Medicine PhD Program, University of Modena and Reggio Emilia, 41100 Modena, Italy

**Keywords:** single-shade composites, color measurement, artificial aging, polymerization, staining agents, spectrophotometer

## Abstract

Composite resins are the material of choice for direct restorations, and their success depends mainly on their color stability, since discoloration causes color mismatch, and consequent patient dissatisfaction. A single- and a multi-shade resin were compared in order to evaluate their pigmentation after immersion in staining substances and to investigate the effect of the polymerization time on their color stability. Two-hundred-and-forty composite specimens were created, half made of a single-shade (Group ONE, n = 120) and half of a multi-shade composite (Group OXP, n = 120). Each group was further divided into ONE30 (n = 60) and OXP30 (n = 60), polymerized for 30″, and ONE80 (n = 60) and OXP80 (n = 60), polymerized for 80″. Randomly, the specimens were immersed in turmeric solution, soy sauce, energy drink, or artificial saliva. By means of a spectrophotometer, ΔE_00_ and WI_d_ were calculated at 24 h (T_0_), at 7 (T_1_), and 30 (T_2_) days. Single-shade composites showed statistically significant differences in color change from the turmeric solution, energy drink, and soy sauce than the multi-shade composites (*p* < 0.005), showing a higher discoloration potential. The polymerization time did not have significative effects on color stability. Single-shade composites showed more color change than multi-shade systems after immersion in staining substances, and the curing time did not influence color variations.

## 1. Introduction

Tooth color is determined by the ways taken by light inside the tooth defined by light scattering, and the absorption of light along these paths [1]. It has been clinically observed that, after placement, some dental materials take on the color appearance of surrounding hard dental tissues, thus improving esthetics. This optical phenomenon is called Blending Effect (BE) or “chameleon effect” [2]. 

Single-shaded composites are single-mass resins developed to adapt chromatically to the adjacent tooth structures, irrespective of the shade of the tooth to be restored, since they are capable of simulating all the shades of the teeth color on which they are applied, using one unique mass [3]. These materials have been developed to make easier restorative procedures by eliminating the color selection phase, reducing the need to use different masses of composite and with different translucencies, and therefore expediting the restorative treatment [4]. It appears that these composites are able to match all shades of the VITA classic scale (A1-D4) by reflecting a certain tooth color wavelength [5], and are very promising for use in clinical practice, particularly when compared to traditional multi-shaded composite systems which require several masses of different colors to make a restoration [6].

Multi-shade conventional composites instead present several opacities and hues that mimic the translucency and chromaticity of dentin and enamel [7]. In order to replicate the optical features of a natural tooth, reconstruction procedures establish the use of several composite layers with different chromas and opacities [8,9].

The pigmentation of resin materials can be due to intrinsic and extrinsic causes [10]. The former cause pigmentation of the composite resin by altering the filler–matrix interface and the matrix itself [11], and are mainly associated with the characteristics of fillers, content of organic components, and polymerization degree [12]. Extrinsic factors instead stem from the extended exposure of the restorations to the oral environment, including inadequate oral hygiene and the absorption of colorants contained in beverages and foods [12]. These are typically superficial and are linked to eating habits, smoking, and oral hygiene. Coffee, red wine, and turmeric are considered the foods with the greatest pigmenting capability, followed by mate and tea [13,14].

To obtain great aesthetic composite restorations, it is mandatory that tooth-colored materials preserve resistance to surface staining and intrinsic color stability [15]. Previous studies have demonstrated that traditional composite materials are susceptible to staining in several staining media [15,16]. Multiple in vitro studies have evaluated the pigmentation of composite restorations following exposure, more or less prolonged, to daily drinks [17,18]. 

Another important factor influencing the esthetic properties and durability of resin composites is the curing efficacy, in other words, the amount of resin that can be converted from monomers to polymers [19]. Polymerization efficiency is mandatory to achieve higher degrees of conversion leading to smaller amounts of residual monomers, which are responsible for the color deterioration [20]. The polymerization depth affects the composite resin’s color, translucency, and application thickness, and the distance from the curing device to the material, time after irradiation, filler size, and dispersion [21]. Insufficient light-curing is significantly associated with a decrease in the physical properties of resin composites [22]; insufficient polymerization can make the matrix easily absorb beverages [21] whereas a more efficient polymerization increases resistance to wear and fracture and improves composite resin color stability [23]. 

In this research project, a single-shade composite resin and a traditional micro-hybrid multi-shade resin were studied, in order to evaluate the pigmentation found following immersion in coloring food substances and the effect of the polymerization time on the color modification of composite resins.

The aim of this in vitro study is to investigate the color differences of a single-shade composite following immersion in staining edible solutions, comparing it to that observed with a traditional multi-shade composite. The effect of the polymerization time of the composites on the color stability was also evaluated.

Particularly, the null hypotheses were that (1) no differences in color variations exist between the two materials, and that (2) curing time does not influence the color stability of the tested materials. 

## 2. Materials and Methods

A total of 240 resin composite blocks (14 mm × 5 mm × 4 mm) were created using a silicone mold (AGG3549 Flat Embedding Moulds, Agar Scientific, Stansted, UK) to achieve identical samples. Half of them were made of a micro-hybrid single-shade composite resin (ONEshade, OliDent, Poznan, Poland) (Group ONE, n = 120), and the other half were made of a micro-hybrid multi-shade composite (OlicoXP A2, OliDent, Poznan, Poland) (Group OXP, n = 120). Each group was further divided into two sub-groups: groups ONE30 (n = 60) and OXP30 (n = 60), polymerized for 30″, and groups ONE80 (n = 60) and OXP80 (n = 60), polymerized for 80″. 

The polymerization time recommended by the manufacturer was 30″, and the descriptions of the tested materials are presented in Table 1.

All the samples were realized through several increments of material (2 mm each), each of which was polymerized in contact with the composite surface using a blue-led medium intensity lamp (Starlight Pro, Mectron, Carasco, Genoa, Italy) at 1400 mW/cm^2^. Before the last curing phase, a Mylar sheet (Mylar, DuPont, Wilmington, DE, USA) was applied on top of the composite to achieve a perfectly glossy and smooth surface with no roughness. Specimens’ layers were cured for 30 or 80 s depending on their subgroup. Subsequently, samples were stored inside a dark container for 24 h in distilled water at 37 °C in order to achieve rehydration. Specimens were then air-dried gently and color coordinates were recorded at T_0_ using an intraoral spectrophotometer (VITA Easyshade V, VITA Zahnfabrik, Bad Säckingen, Germany), by positioning the tip perpendicular to the specimens’ surface with natural daylight and a grey background [24]. 

The following CIELAB color coordinates were recorded: L* (lightness), a* (green-red coordinate), b* (blue-yellow coordinate), C* (chroma), and h° (hue). Measurements were taken in the middle of the sample surface (three measurements per unit) after 24 h of immersion in distilled water (T_0_) [25]. After randomization, the composite blocks were then divided into 16 groups (n = 15): ONE30, ONE80, OXP30, and OXP80 specimens were divided to be immersed in 4 different colorant substances.

The following substances were selected for immersion of the samples: turmeric powder–water solution, soy sauce, energy drink, and artificial saliva (Table 2).

Before immersing the samples in the drink formulations, the solution with turmeric was prepared as indicated in Table 3. The other substances were introduced in their pure form into individual, hermetically sealable plastic containers (Fisherbrand™ Screw Top, Fisher Scientific, Hampton, NH, USA). 

The drink formulations were replaced every 24 h for 30 days, during which the samples were kept in incubation at 37 °C in a box away from light. The measurements with the spectrophotometer were carried out against a grey background and natural light [24], calibrating the instrument after each 3 measurements [28], at T_0_ (after 24 h in distilled water at 37 °C), at 7 days (T_1_), and 30 days (T_2_) [25].

From the data obtained with the spectrophotometer, the ΔE_00_ and WI_d_ values were calculated:$${\Delta E}_{00}=\sqrt{{\left(\frac{\Delta L^{\prime }}{K_{L}S_{L}}\right)}^{2}+{\left(\frac{\Delta C^{\prime }}{K_{C}S_{C}}\right)}^{2}+{\left(\frac{\Delta H^{\prime }}{K_{H}S_{H}}\right)}^{2}+R_{T}\left(\frac{\Delta C^{\prime }}{K_{C}S_{C}}\right)\left(\frac{\Delta H^{\prime }}{K_{H}S_{H}}\right)}\;{\mathrm{WI}}_{d}=0.511L*\;-\;2.324a*\;-\;1.100b*$$

Before each measurement, the samples were extracted from the solution, rinsed thoroughly under running water, and dried with a paper towel [29].

### Statistical Analysis

Statistical analysis was performed using the STATA program version 17 (StataCorp LP 4905 Lakeway Drive, College Station, TX 77845, USA). According to the RoBDEMAT risk of bias tool for pre-clinical dental research [30], the sample size was calculated depending on a previous study as a reference [31]. According to this research, the response within each subject group was normally distributed with a standard deviation of 1.4. If the actual difference in the experimental and control means is 1.8, the study minimally needed 11 subjects in each group to reject the null hypothesis that the population means of the experimental and control groups are equal with probability (power) 0.8. The type I error probability associated with this null hypothesis test is 0.05. The total sample size increased to 15 subjects per group to compensate for 20% dropout.

Means and standard deviations were used to summarize the data. Data from color coordinates (CIE L*, a*, b*, C*, and h°) were statistically analyzed using analysis of variance (one-way ANOVA) and Tukey’s multiple comparison test with Bonferroni correction. One-way ANOVA was performed to compare the effects of the color difference ΔE_00_ values among the materials and whiteness index differences (ΔW_Id_). Paired *t*-tests were used to compare continuous measures between groups. A *p*-value of <0.05 was considered statistically significant.

## 3. Results

Mean color differences (ΔE_00_) and standard deviations (SDs) of resin specimens (ONE and OXP) between T_0_ (baseline) and the different follow-ups (T_1_: 7 days, T_2_: 30 days), after immersion in various media (artificial saliva, turmeric powder–water solution, energy drink, soy sauce) and at different curing times (30″ and 80″) are presented in Table 3.

Statistically significant differences were observed between OXP_80 and ONE_80 at T_1_ for turmeric (*p* < 0.001) and energy drink (*p* = 0.005); at T_2_ for artificial saliva (*p* = 0.039) and energy drink (*p* = 0.001).

Statistically significant differences were observed between OXP_30 and ONE_30 at T_1_ for energy drink (*p* = 0.004); at T_2_ for energy drink (*p* = 0.001) and soy (*p* < 0.001). 

Statistically significant differences were observed between ONE_80 and ONE_30 at T_1_ for turmeric (*p* = 0.004). 

Statistically significant differences were observed between OXP_80 and OXP_30 at T_1_ for turmeric (*p* < 0.001); at T_2_ for artificial saliva (*p* = 0.005).

Considering ΔE_00_(T_0_–T_1_) and ΔE_00_(T_0_–T_2_), statistically significant differences were observed in ONE_80 specimens for turmeric (*p* = 0.006), energy drink (*p* < 0.001), and soy (*p* < 0.001); in OXP_30 specimens for turmeric (*p* = 0.006), energy drink (*p* < 0.001), and soy (*p* < 0.001); in OXP_80 specimens for artificial saliva (*p* = 0.002) and soy (*p* < 0.001); in ONE_30 specimens for turmeric (*p* < 0.001), energy drink (*p* < 0.001), and soy (*p* < 0.001).

The linear prediction values of ΔE_00_ at baseline and T_2_ are shown in Figure 1.

Mean whiteness index differences (ΔW_Id_) and standard deviations (SDs) of resin blocks (ONE and OXP) between T_0_ (baseline) and the different follow-ups (T_1_: 7 days, T_2_: 30 days), after immersion in various media (artificial saliva, turmeric powder–water solution, energy drink beverage, soy sauce) and at different curing times (30″ and 80″) are presented in Table 4.

Statistically significant differences were observed between ONE_80 and OXP_80 at T_0_ for artificial saliva (*p* < 0.001), turmeric (*p* = 0.025), energy drink (*p* < 0.001), and soy (*p* < 0.001); at T_1_ for artificial saliva (*p* < 0.001), turmeric (*p* < 0.001), energy drink (*p* = 0.013), and soy (*p* = 0.011); at T_2_ for artificial saliva (*p* < 0.001). 

Statistically significant differences were observed between ONE_30 and OXP_30 at T_0_ for artificial saliva (*p* < 0.001), turmeric (*p* < 0.001), and energy drink (*p* = 0.001); at T_1_ for artificial saliva (*p* < 0.001), turmeric (*p* < 0.001), and energy drink (*p* = 0.001); at T_2_ for artificial saliva (*p* < 0.001), turmeric (*p* = 0.015), and soy (*p* < 0.001). 

Statistically significant differences were observed between ONE_80 and ONE_30 at T_0_ for artificial saliva (*p* < 0.001), turmeric (*p* = 0.003), energy drink (*p* = 0.015), and soy (*p* = 0.001); at T_1_ for artificial saliva (*p* < 0.001) and soy (*p* < 0.001); at T_2_ for soy (*p* = 0.025). 

Statistically significant differences were observed between OXP_80 and OXP_30 at T_0_ for artificial saliva (*p* < 0.001), turmeric (*p* = 0.031), energy drink (*p* < 0.001), and soy (*p* < 0.001); at T_1_ for artificial saliva (*p* < 0.001), turmeric (*p* < 0.001), and soy (*p* = 0.027), at T_2_ for artificial saliva (*p* < 0.001).

Linear prediction of WI_D_ at baseline, T_1_, and T_2_ are shown in Figure 2.

## 4. Discussion

From the achieved results, it can be stated that the null hypotheses formulated were partially rejected, due to the following:
–There were differences in color variations between the two materials;–Curing time does not influence the color stability of the tested materials.

The primary purpose of this in vitro study was to analyze the color differences of a single-shade composite following immersion in pigmented food solutions, comparing them to the variations observed with a traditional multi-shade composite. Overall, the results showed that the surfaces of all tested samples underwent a certain amount of color change.

Although there have been scientific publications on the BE of single-shade resin composites [28,32], studies on the color stability of these materials are quite limited [12,25,27,33,34].

A recent study highlighted how these composites were able to show a higher chromatic change compared to traditional multi-shade resins, after immersion in red wine, coffee, and black tea [25].

Similarly, Chen et al. evaluated the color stability of a single-shade composite compared to three multi-shade resins, after immersion in coffee [12]. Although all materials showed different color change degrees, the greatest pigmentation was expressed by the single-shade composite.

Another research group studied the color stability of a single-shade composite compared to a multi-shade one, after immersion in tea and red wine. Although both materials showed a high degree of pigmentation, the color variation of the multi-shade composite was significantly lower than the single-shade composite [27].

Ebaya et al. evaluated and compared the color stability of two single-shade composite restorations after storing them in different staining media (black tea and cola). They concluded that the aging procedure exerted a negative effect on single-shade composite color stability [35].

To our knowledge, no studies have been conducted yet evaluating the color variations of single-shade composites after immersion in the substances tested in the present study: a solution of water and turmeric powder, an orange-colored energy drink, and soy sauce.

Specimens immersed in the water/turmeric solution exhibited the greatest color change, regardless of the type of composite. This result is consistent with previously published studies, where a turmeric-based solution caused the most significant pigmentation in nano-filled and micro-hybrid traditional composite resin samples [26], and high ΔE values in all tested composites, regardless of the initial shade and the composition of the resin matrix [36]. 

It has been suggested that the absorption of yellow color by these restorative materials could be attributed to the compatibility of pigments with the composites’ organic matrix [37]. Furthermore, it has been demonstrated that the pigmented compounds responsible for the color of spices contain polyphenolic structures, and polyphenols are implicated as causal agents in dental pigmentation [38]. 

It is known that soy sauce contains caramel hydrophilic coloring pigment. Color alterations and microhardness variations of a micro-hybrid composite were evaluated after immersion in various colored foods, investigating the possible correlation between these two variables. According to these authors, the absorption by the hydrophilic component of the organic matrix, especially the characteristic of Bis-GMA, could be responsible for the pigmentation of the composite resin. The ΔE values after 28 days were considered unacceptable from an esthetic point of view, but the greatest color variations in the aforementioned study were caused, in order, by mustard and ketchup [39].

In the present study, soy sauce showed a lower staining capability compared to turmeric in both types of composites investigated, with a statistically significant difference in pigmentation. The samples immersed in the water/turmeric solution and soy sauce showed an incremental color change with increasing immersion time [26].

Regarding the Monster^®^ “Pacific Punch” energy drink, it was shown that there was a pigmentation increase from T_0_ to T_2_ for the groups ONE30, ONE80, and OXP30, but no statistically significant color variation was recorded for the OXP80 group from T_1_ to T_2_.

Soft drinks have been widely used to evaluate composite color stability [40], but only a few were tested on single-shade composites [35,41].

In a recent publication, the authors evaluated the color stability of two composite resins after exposure to beverages such as Cola, Pepsi, and Red Bull after 15 days. In all specimens, the tested composites discolored less in the Red Bull energy drink [41]. 

A recent publication compared the color stability of micro-hybrid composites and nanocomposites after exposure to common soft drinks among adolescents. All soft drinks resulted in clinically unacceptable discoloration of the composite materials with maximum discoloration occurring following the immersion of composite materials in iced tea [40].

De Alencar et al. evaluated the color of nanoparticle and nanohybrid composites after immersion in açaí juice and grape juice. Grape juice produced the greatest color change in nanocomposites after 2 weeks, and açaí juice made the color unacceptable clinically only after 12 weeks [33].

Berber et al. evaluated the color stability of resin composites using different drinks. The authors concluded that when drinks with and without sugar were compared, all groups with sugar demonstrated a higher color difference than without sugar [42].

Unexpectedly, specimens from the OXP80 group immersed in artificial saliva showed a slight color change. Artificial saliva does not contain pigments, and therefore the color change could be due to water absorption of the matrix. This could cause swelling and plasticization of the polymer together with the formation of interstitial spaces between the filler and the resinous matrix, thus allowing the color variation [43]. However, from a clinical point of view, the instrumental color measurements carried out with the spectrophotometer at T_0_ and T_2_ did not highlight evident chromatic variations when based on the Vita Classical reference scale.

The secondary goal of this research project was to evaluate if different composite polymerization timings could affect resin color, as half of the samples were polymerized for 30″ and the other half for 80″. The results showed that all specimens, regardless of the curing time, underwent a certain amount of color change.

This lack of significant differences was found also by Gonder and Fidan in 2022. The authors aimed to evaluate the effect of different polymerization times on color change, translucency parameter, and surface hardness of resin composites after thermocycling. Different curing times did not affect the color change of the composite materials [21].

It is interesting to note that the ΔW_Id_ showed a greater initial whiteness for the specimens cured for 80″, in both groups. This could indicate that a longer curing time could have an effect on the initial composites’ whiteness.

It is mandatory to highlight that the present research has been conducted in vitro and that the interpretation of the results could present some limitations. 

First, the specimens were always in contact with the full concentration of the staining substances. Accordingly, the role of saliva was not simulated. 

Second, the specimens were soaked in the food substances at 37 °C, but usually some drinks could be consumed cold. Therefore, the amount of discoloration observed in the present study could have been influenced more than what should occur when the immersion is conducted at low temperatures. 

An evident limitation of this research protocol could be represented by the choice of the spectrophotometer used for this study. The VITA Easyshade V is a clinical device that has been used in several studies and illuminates the tooth with a 6500 K light [6,12,24,28,32,33,40,42]. Even though a bench spectrophotometer could represent the most suitable instrument for an in vitro study [44], based on recently published data [12,24,28,32,33,40,45], a clinical spectrophotometer was chosen for the present study in order to reproduce a clinical situation in the most precise way. A study by Dozić et al. found VITA Easyshade to be, in vitro and in vivo, the most precise among five other similar devices, providing good reliability and accuracy [46]. More recently, another publication evaluated the reliability of two clinically applicable spectrophotometers under laboratory and clinical conditions, concluding that both spectrophotometers produced reliable and accurate measurements and can therefore be recommended for clinical and laboratory determinations of tooth color [47].

Additionally, no thermocycling was considered in the present research. Composites aging in the oral cavity and tooth-brushing could modify the extent of discoloration from these staining substances. 

Another limit of this research protocol could be that both tested resins are micro-hybrid composites. First-generation conventional composites presented with average particle sizes that exceeded 1 μm. These macro-filled materials were very resistant but difficult to polish and impossible to retain surface smoothness. In order to address the important issue of long-term esthetics, which was lacking with the macro-filled composites, manufacturers began to formulate micro-filled materials, until the most recent development of the nano-filled composites, containing only nanoscale particles [48]. For this reason, significant improvements in surface smoothness/polish retention have been reported for nano-filled compared to conventional micro-filled materials [49]. The latest generation of micro-hybrid composites consist of both microscale and nanoscale (∼0.02 micron) glass fillers [49]. It may appear that composites featuring smaller filler particles, such as nano-hybrid composite resins, could potentially provide greater color stability. In the literature, however, there is scientific evidence relating to micro-hybrid composite resins that show higher color stability than nano-hybrid ones [41]. A research group, for example, conducted a study to examine the impact of different soft drinks (iced tea, sports drinks, orange juice, and cola) on micro- and nano-hybrid composites. The results indicated that the micro-hybrid composites demonstrated superior color stability after immersion in all the drinks tested [40]. Further studies have shown how increasing the size of the filler particles can lead to a reduction in the pigmentation of the composite, probably due to the lower matrix–filler ratio [33,41].

Therefore, it could be of interest to evaluate the resistance to discoloration of the restorations with thermocycling incorporated, and to establish new in vitro studies that consider the effect of the assessed staining food substances, perhaps in diluted forms, on extracted natural teeth restored with the tested composites.

## 5. Conclusions

According to the results of this in vitro study in which the color stability of two different resin composites were examined, the following conclusions were obtained:
Single-shade resin composites in turmeric solution, energy drink, and soy sauce showed more color changes than multi-shade systems;The whiteness index decreased over time, regardless of the type of composite and the staining substance;The curing time did not influence the color variations of both composite groups.

Single-shade composites seem to be more susceptible to discoloration, which may affect their long-term success rate, although the potential discoloration might be limited by dietary adjustments. Patients could be asked about their diet habits and made aware of their potential to discolor composite restorations. At the same time, dentists have to remember the discoloration capability of composite resin restorations.

## Figures and Tables

**Figure 1 dentistry-12-00285-f001:**
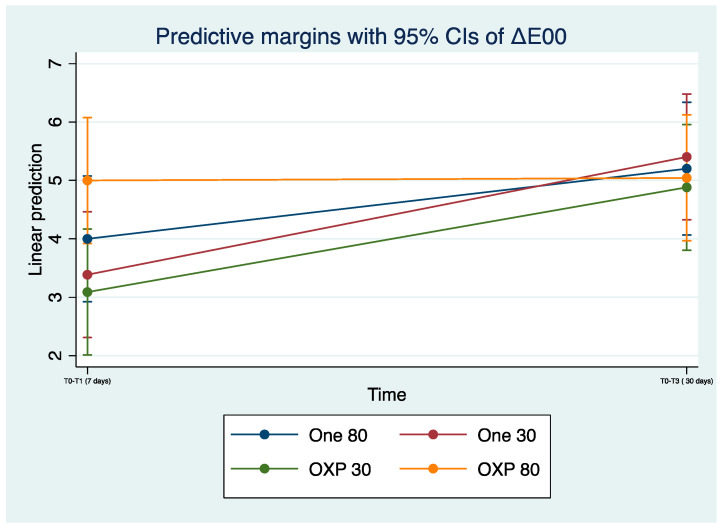
Linear prediction of ΔE_00_. Means (dots), 95% confidence interval (whiskers), samples at baseline T_0_ (time T_0_–T_1_) and at T_2_ (time T_0_–T_3_).

**Figure 2 dentistry-12-00285-f002:**
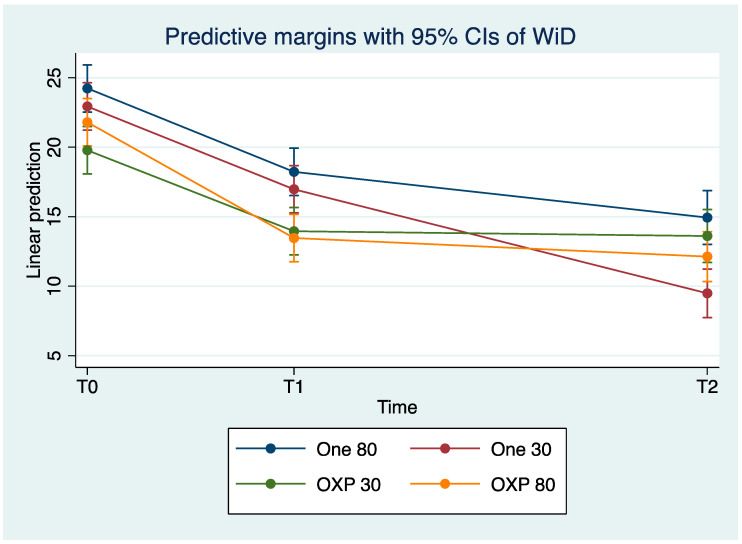
Linear prediction of WI_D_. Means (dots), 95% confidence interval (whiskers), samples at baseline T_0_, T_1_, at T_2_.

**Table 1 dentistry-12-00285-t001:** Details of the tested composites.

Composite	Filler Particles	Resin Matrix	Filler	Manufacturer	Lot #
ONEshade	Micro-hybrid	Glass powder, polyurethane dimethacrylate, silicon dioxide, Bis-GMA, tetramethylene dimethacrylate	75% weight, 53% volume. Inorganic filler particles (0.005–3.0 μm)	Olident (Podłęże, Polonia)	2023004852
Olico XP	Micro-hybrid	Glass powder, dimethacrylates, silicon dioxide, Bis-GMA	80% weight, 68% volume. Inorganic filler particles (0.05–0.9 µm)	Olident (Podłęże, Polonia)	52301159C

**Table 2 dentistry-12-00285-t002:** Description of the food substances used and composition.

LIQUID	CONTENT
Turmeric powder–water solution	Obtained by diluting 1 g of turmeric powder in 1 L of distilled water and boiling for 10 min to obtain a standardized concentration [26].
Soy sauce (Kikkoman^®^, Kikkoman Trading Europe GmbH, Düsseldorf, Germany)	Water, soy, wheat, salt.
Energy drink (Monster Pacific Punch^®^, Monster Energy, Corona, CA, USA)	Carbonated water, sugar, glucose, orange concentrate, apple concentrate, taurine, acidity regulators E330–E331, raspberry concentrate, goyava puree, cherry concentrate, preservatives E202–E211, caffeine, natural flavourings, vitamins B3–B6-B2–B12, sweetener E955, pineapple concentrate, passion fruit concentrate, thickener E414, colors E129*–E133, salt, emulsifier E445, inositol, carnitine tartrate. Caffeine: 333 mg/L.
Control	Artificial saliva [27].

**Table 3 dentistry-12-00285-t003:** Mean color differences (ΔE_00_) and standard deviations of resin specimens between baseline and the different follow-ups, after immersion in various media and at different curing times.

Mean ΔE_00_ ± SD						
	**T_0_–T_1_ (7 days)**		***p*-Value**	**T_0_–T_1_ (7 days)**		***p*-Value**
	**ONE80**	**OXP80**		**ONE80**	**ONE30**	
Artificial saliva	0.7 ± 0.2	0.7 ± 0.1	0.676	0.7 ± 0.2	0.8 ± 0.5	0.630
Turmeric	9.4 ± 0.6	14 ± 1.5	<0.001	9.4 ± 0.6	6.6 ± 2.8	0.004
Energy drink	2.1 ± 0.6	1.4 ± 0.5	0.005	2.1 ± 0.6	1.9 ± 0.7	0.612
Soy	3.7 ± 0.4	3.6 ± 0.9	0.642	3.7 ± 0.4	4.2 ± 0.9	0.171
	**T_0_–T_2_ (30 days)**			**T_0_–T_2_ (30 days)**		
	**ONE80**	**OXP80**		**ONE80**	**ONE30**	
Artificial saliva	0.7 ± 0.2	0.8 ± 0.1	0.039	0.7 ± 0.2	1.3 ± 1.6	0.184
Turmeric	17.7 ± 0.1	15.1 ± 2.4	-	17.7 ± 0.1	16.0 ± 1.3	-
Energy drink	3.2 ± 1.0	1.8 ± 0.6	0.001	3.2 ± 1.0	2.8 ± 0.9	0.389
Soy	6.7 ± 1.4	6.5 ± 2.4	0.718	6.7 ± 1.4	7.6 ± 1.1	0.095
	**T_0_–T_1_ (7 days)**			**T_0_–T_1_ (7 days)**		
	**ONE30**	**OXP30**		**OXP30**	**OXP80**	
Artificial saliva	0.8 ± 0.5	0.8 ± 0.7	0.898	0.8 ± 0.7	0.7 ± 0.1	0.071
Turmeric	6.6 ± 2.8	6.6 ± 1.0	0.988	6.6 ± 1.0	14 ± 1.5	<0.001
Energy drink	1.9 ± 0.7	1.2 ± 0.2	0.004	1.2 ± 0.2	1.4 ± 0.5	0.206
Soy	4.2 ± 0.9	3.6 ± 0.7	0.056	3.6 ± 0.7	3.6 ± 0.9	0.967
	**T_0_–T_2_ (30 days)**			**T_0_–T_2_ (30 days)**		
	**ONE30**	**OXP30**		**OXP30**	**OXP80**	
Artificial saliva	1.3 ± 1.6	0.9 ± 0.3	0.459	0.9 ± 0.3	0.8 ± 0.1	0.005
Turmeric	16.0 ± 1.3	12.1 ± 0.1	0.125	12.1 ± 0.1	15.1 ± 2.4	-
Energy drink	2.8 ± 0.9	1.9 ± 0.5	0.001	1.9 ± 0.5	1.8 ± 0.6	0.740
Soy	7.6 ± 1.1	5.3 ± 1.2	<0.001	5.3 ± 1.2	6.5 ± 2.4	0.145

**Table 4 dentistry-12-00285-t004:** Whiteness index (W_Id_) and standard deviations of resin specimens between baseline and the different follow-ups, after immersion in various media and at different curing times.

Witeness Index ± SD												
	**T_0_**		***p*-value**	**T_0_**		***p*-value**	**T_0_**		***p*-value**	**T_0_**		***p*-value**
**ONE80**	**OXP80**		**ONE30**	**OXP30**		**ONE80**	**ONE30**		**OXP30**	**OXP80**	
Artificial saliva	24.7 ± 0.5	22 ± 0.6	<0.001	22.7 ± 1.6	19.7 ± 1.4	<0.001	24.7 ± 0.5	22.7 ± 1.6	<0.001	19.7 ± 1.4	22 ± 0.6	<0.001
Turmeric	23.7 ± 0.7	21.3 ± 3.4	0.015	22.7 ± 1.1	19 ± 0.9	<0.001	23.7 ± 0.7	22.7 ± 1.1	0.003	19 ± 0.9	21.3 ± 3.4	0.021
Energy drink	24.9 ± 0.7	22 ± 0.6	<0.001	23.9 ± 1.5	20.3 ± 0.8	<0.001	24.9 ± 0.7	23.9 ± 1.5	0.015	20.3 ± 0.8	22 ± 0.6	<0.001
Soy	23.7 ± 0.7	21.9 ± 0.5	<0.001	22.5 ± 0.8	20.2 ± 1	<0.001	23.7 ± 0.7	22.5 ± 0.8	0.001	20.2 ± 1	21.9 ± 0.5	<0.001
Total	24.2 ± 0.9	21.8 ± 1.8	<0.001	22.9 ± 1.4	19.8 ± 1.1	<0.001	24.2 ± 0.9	22.9 ± 1.4	<0.001	19.8 ± 1.1	21.8 ± 1.8	<0.001
	**T_1_**			**T_1_**			**T_1_**			**T_1_**		
	**ONE_80**	**OXP_80**		**ONE_30**	**OXP_30**		**ONE_80**	**ONE_30**		**OXP_30**	**OXP_80**	
Artificial saliva	24.8 ± 0.7	21.3 ± 0.5	<0.001	22.6 ± 1.4	18 ± 1.1	<0.001	24.8 ± 0.7	22.6 ± 1.4	<0.001	18 ± 1.1	21.3 ± 0.5	<0.001
Turmeric	11 ± 2.3	−0.6 ± 4.3	<0.001	13.4 ± 2.9	8.1 ± 3.4	<0.001	11 ± 2.3	13.4 ± 2.9	0.056	8.1 ± 3.4	−0.6 ± 4.3	<0.001
Energy drink	20 ± 2.1	18.3 ± 1.6	0.013	19.4 ± 1.8	17.4 ± 1	0.001	20 ± 2.1	19.4 ± 1.8	0.351	17.4 ± 1	18.3 ± 1.6	0.075
Soy	17.1 ± 1.3	14.8 ± 2.9	0.011	12.5 ± 2.4	12.5 ± 2	0.947	17.1 ± 1.3	12.5 ± 2.4	<0.001	12.5 ± 2	14.8 ± 2.9	0.027
Total	18.2 ± 5.3	13.5 ± 8.9	<0.001	17 ± 4.8	14 ± 4.5	<0.001	18.2 ± 5.3	17 ± 4.8	0.177	14 ± 4.5	13.5 ± 8.9	0.703
	**T_2_**			**T_2_**			**T_2_**			**T_2_**		
	**ONE_80**	**OXP_80**		**ONE_30**	**OXP_30**		**ONE_80**	**ONE_30**		**OXP_30**	**OXP_80**	
Artificial saliva	23.2 ± 0.6	21.2 ± 0.9	<0.001	22.5 ± 1.5	18.9 ± 0.9	<0.001	23.2 ± 0.6	22.5 ± 1.5	0.108	18.9 ± 0.9	21.2 ± 0.9	0.001
Turmeric	−18 ± 2.1	−6.9 ± 5.5	-	−10.9 ± 6.4	−6.3 ± 2.1	0.015	−18 ± 2.1	−10.9 ± 6.4	-	−6.3 ± 2.1	−6.9 ± 5.5	-
Energy drink	17,2 ± 3	17.1 ± 2.1	0.958	16.6 ± 2.6	15.7 ± 1.6	0.191	17.2 ± 3	16.6 ± 2.6	0.675	15.7 ± 1.6	17.1 ± 2.1	0.058
Soy	8.9 ± 3.4	9.5 ± 6.1	0.768	5.6 ± 3.3	10.2 ± 2.4	<0.001	8.9 ± 3.4	5.6 ± 3.3	0.025	10.2 ± 2.4	9.5 ± 6.1	0.722
Total	14.9 ± 9.5	12.1 ± 10.5	0.161	9.5 ± 12.9	13.6 ± 6.5	0.046	14.9 ± 9.5	9.5 ± 12.9	0.017	13.6 ± 6.5	12.1 ± 10.5	0.398

TOTAL values stand for mean WI_D_ data with no distinction of media.

## Data Availability

Data supporting the reported results can be obtained, on request, by writing to the corresponding author.

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
