# Peer review of "Color Stability Assessment of Single- and Multi-Shade Composites Following Immersion in Staining Food Substances"

_dentistry, 2024, doi:10.3390/dj12090285_

Round 1
Reviewer 1 Report
Comments and Suggestions for Authors
The paper is very interesting and unique. The authors are requested to address the following points to improve the quality of this manuscript:
- Similarity index is too high. Authors are required to perform significant paraphrasing.
- The abstract should start with opening (background) statement that reflect the importance of this study and the outstanding research question.
- The introdution was poorly structured. Please make it more concise and focused. Please update references since some of them are old.
- Authors may cite this research for relevance: Gupta S, Sayed ME, Gupta B, Patel A, Mattoo K, Alotaibi NT, Alnemi SI, Jokhadar HF, Mashhor BM, Othman MA, Mugri MH, Porwal A, Patil S. Comparison of Composite Resin (Duo-Shade) Shade Guide with Vita Ceramic Shades Before and After Chemical and Autoclave Sterilization. Med Sci Monit. 2023 Jun 30;29:e940949. doi: 10.12659/MSM.940949. PMID: 37386761; PMCID: PMC10318931.
- How did authors determine sample size? please elaborate on this point.
- Standard curing distance from samples should be mentioned.
- Specifications of the samples' fabrication mold should be added in the text.
- Authors may add graphs to explain changes over time and difference between staining media.
- Limitations and directions for future research should be added.
- Clinical relevance and recommendations should be added to the conclusion section.
- Conclusion section is concise and well-structured.
Author Response
The paper is very interesting and unique. The authors are requested to address the following points to improve the quality of this manuscript:
Comment 1: Similarity index is too high. Authors are required to perform significant paraphrasing.
Response: Thank you for your suggestion. Introduction, Methods and Discussion sections were modified significantly.
Comment 2: The abstract should start with opening (background) statement that reflect the importance of this study and the outstanding research question.
Response: Thank you for your suggestion. We added a short background statement in order to not exceed the maximum number of words allowed (Lines 14-16).
Comment 3: The introduction was poorly structured. Please make it more concise and focused. Please update references since some of them are old.
Response: Thank you for your request. We modified the Introduction section in order to be more concise and focused on the tested subjects. References were updated as requested (Ref. #2 and #9).
Comment 4: Authors may cite this research for relevance: Gupta S, Sayed ME, Gupta B, Patel A, Mattoo K, Alotaibi NT, Alnemi SI, Jokhadar HF, Mashhor BM, Othman MA, Mugri MH, Porwal A, Patil S. Comparison of Composite Resin (Duo-Shade) Shade Guide with Vita Ceramic Shades Before and After Chemical and Autoclave Sterilization. Med Sci Monit. 2023 Jun 30;29:e940949. doi: 10.12659/MSM.940949. PMID: 37386761; PMCID: PMC10318931.
Response: Thank you for your suggestion. We added this research for relevance (Ref. #9).
Comment 5: How did authors determine sample size? Please elaborate on this point.
Response: Thank you for this question. At the end of 2022, a study group formed by members of European Federation of Conservative Dentistry and members of the Dental Materials Group of the International Association for Dental Research developed a risk of bias tool for pre-clinical dental materials research studies (doi: 10.1016/j.jdent.2022.104350). According to this RoBDEMAT risk of bias tool for pre-clinical dental research, one of the accepted ways to determine the sample size is to calculate it depending on a previous study. Therefore, we chose a previously published study to determine our specimens sample size.
We added this explanation (Lines: 140-142).
Comment 6: Standard curing distance from samples should be mentioned.
Response: Thank you for your indication. Each composite layer was cured in contact using a blue-led medium intensity lamp. We added this explanation (Lines 102-104).
Comment 7: Specifications of the samples' fabrication mold should be added in the text.
Response: As requested, we added the samples’ fabrication mold specifications (Line: 92).
Comment 8: Authors may add graphs to explain changes over time and difference between staining media.
Response: Thank you for the feedback. To enhance the clarity and understanding of the data presented we incorporated graphs to illustrate changes over time and differences between staining media (Lines 178 and 205). These visual aids will complement the revised tables by providing a clearer depiction of trends and comparative data. These additions aim to improve the overall presentation and make the data more comprehensible.
Comment 9: Limitations and directions for future research should be added.
Response: Thank you for your suggestions. We added limitations to the study (Lines: 301-344) and indications for future research (Lines: 345-348) in the Discussion section.
Comment 10: Conclusion section is concise and well-structured. Clinical relevance and recommendations should be added to the conclusion section.
Response: Thank you for the suggestion. We added clinical relevance and recommendations as requested (Lines 357-361).
Reviewer 2 Report
Comments and Suggestions for Authors
-Introduction
- Materials and Methods
All Tables presented must be correctly formatted.
Line 112: cm2 must be corrected to cm2.
Why was Vita Easychade chosen for the study? Didn't benchtop spectrophotometers allow for more accurate readings?
Lines 131 and 132: Repeated text!
Line 135: what type of sealable plastic containers?
Line 163: ≤0.05 must be corrected to <0.05.
-Results
Very confusing presentation! I suggest reviewing the entire section.
It is not very reasonable to have so much information as a footer in the Tables. There are friendlier ways to indicate statistical differences.
-Discussion
The paragraph at lines 359-362 should start the Discussion.
Next, each of the null hypotheses must be discussed and compared with the literature in the area.
Line 259: micr-ohybrid must be corrected to micro-hybrid.
Line 264: "Some authors" and cite the reference #38 only.
Lines 334-335: acai must corrected to açaí.
What are the limitations of this study?
What suggestions do you make for future studies?
What is the major contribution they make and how can these results be translated into clinical practice?
Comments on the Quality of English Language
Minor corrections are needed.
Author Response
Comment 1: All Tables presented must be correctly formatted.
Response: Thank you for your indication. All tables have been correctly formatted.
Comment 2: Line 112: cm2 must be corrected to cm2.
Response: Thank you, correction has been made (Line 104).
Comment 3: Why was Vita Easyshade chosen for the study? Didn't benchtop spectrophotometers allow for more accurate readings?
Response: Thank you for highlighting this important point. VITA Easyshade V is a clinical device that has been used in several in vitro studies. Even though a bench spectrophotometer could represent the most suitable instrument for a pre-clinical study, a clinical spectrophotometer was chosen to reproduce in the most precise way a clinical situation. Moreover, this device was found to be, both in vitro and in vivo, the most precise among five other similar devices [References #46,47].
We added this point as one of the limitations of our study (Lines: 309-321).
Comment 4: Lines 131 and 132: Repeated text!
Response: Thank you for this observation. We removed the repeated text.
Comment 5: Line 135: what type of sealable plastic containers?
Response: Thank you for the questions. We specified the type of containers (Lines: 125-126).
Comment 6: Line 163: ≤0.05 must be corrected to <0.05.
Response: Thank you, correction has been made.
Comment 7: Very confusing presentation! I suggest reviewing the entire section. It is not very reasonable to have so much information as a footer in the Tables. There are friendlier ways to indicate statistical differences.
Response: Thank you for the feedback. We understand that the current presentation of Tables 3 and 4, particularly with extensive information in the footers, is confusing. To improve clarity and readability we reviewed the entire section to streamline the presentation of data. This will involve restructuring the tables to present information more clearly and concisely.
Enhanced Statistical Indicators: Instead of placing a large amount of information in the footers, we used more intuitive methods to indicate statistical differences. This may include the new column in the table to denote statistical significance.
These changes are aimed at making the tables more user-friendly and the statistical differences easier to interpret.
Comment 8: The paragraph at lines 359-362 should start the Discussion.
Response: We moved the sentence at the beginning of the Discussion section (Lines 210-213).
Comment 9: Next, each of the null hypotheses must be discussed and compared with the literature in the area.
Response: Thank you for these indications. We modified the Discussion section as suggested, discussing the two null-hypotheses after the first paragraph (Starting from lines: 214 and 290).
Comment 10: Line 259: micr-ohybrid must be corrected to micro-hybrid.
Response: Done.
Comment 11: Line 264: "Some authors" and cite the reference #38 only.
Response: Thank you for your note. We replaced “Some authors” with “A Research group”.
Comment 12: Lines 334-335: acai must correct to açaí.
Response: Done.
Comment 13: What are the limitations of this study?
Response: Thank you for your suggestions. We added limitations to the study in the Discussion section (Lines: 301-344).
Comment 14: What suggestions do you make for future studies?
Response: Thank you for this question. We added suggestions for future studies in lines 345-348.
Comment 15: What is the major contribution they make and how can these results be translated into clinical practice?
Response: Thank you for the questions. We added clinical relevance and recommendations in the Conclusion section (Lines 357-361).
Reviewer 3 Report
Comments and Suggestions for Authors
I congratulate the authors for their work.
I have some recommendations:
1) Title
If possible, simplify or reduce the title. For instance, " Color stability of single- and multi-shade composites following immersion in staining food substances " OR " Color stability assessment of single- and multi-shade composites following immersion in staining food substances ".
2) Materials and methods
- I would suggest to remove Table 2 as the information is already explained
- Is reference [26] a standard Reference point? Is it necessary to use it for the information in the following text: "...roughness [26]. Specimens’ layers were cured for 30 or 80 seconds depending on their subgroup. Subsequently, samples were stored in distilled water at 37°C for 24 hours to achieve rehydration [26] inside a dark container. Specimens were then gently air-dried and color coordinates were recorded at T0 using an intraoral spectrophotometer (VITA Easyshade V, VITA Zahnfabrik, Bad Säckingen, Germany), by placing the tip perpendicular to the sample surfaces with a grey background and natural daylight [26]." [lines 114-121]. Please offer an explanation.
- Same question for References 30 and 31[lines 139-141]. Do they represent a standard point necessary to be used in the Materials and methods?
3) Discussions
-please check row 277 => there is Reference 41 "[14,27,29,39,41]" and then row 292 -> there is Reference 40 "color stability [40]"
- I would suggest revising and rephrasing the following statement: "One of the few articles in which the pigmentation effect of energy drinks has been evaluated on composites was published in 2024" [lines 324-325].
A simple search on Pubmed with the following search words: "energy drink AND colour AND composites resin" will show the information from 2012 at least (Erdemir U, Yildiz E, Eren MM. Effects of sports drinks on color stability of nanofilled and microhybrid composites after long-term immersion. J Dent. 2012 Dec;40 Suppl 2:e55-63. doi: 10.1016/j.jdent.2012.06.002. Epub 2012 Jun 17. PMID: 22713737). See also other related articles in the Pubmed search list.
4) References
-please check Ref 40 (row 466)
-please check all the References' styles to be according to the Journal's recommendations
Author Response
Comment 1: I congratulate the authors for their work.
Response: Thank you for your opinion and for all the useful indications.
Comment 2: Title. If possible, simplify or reduce the title. For instance, " Color stability of single- and multi-shade composites following immersion in staining food substances " OR " Color stability assessment of single- and multi-shade composites following immersion in staining food substances".
Response: Thank you for this suggestion. We modified the title as requested (Lines 2-3).
Comment 3: Materials and methods. I would suggest to remove Table 2 as the information is already explained
Response: Thank you for your advice. We removed Table 2.
Comment 4: Is reference [26] a standard Reference point? Is it necessary to use it for the information in the following text: "...roughness [26]. Specimens’ layers were cured for 30 or 80 seconds depending on their subgroup. Subsequently, samples were stored in distilled water at 37°C for 24 hours to achieve rehydration [26] inside a dark container. Specimens were then gently air-dried and color coordinates were recorded at T0 using an intraoral spectrophotometer (VITA Easyshade V, VITA Zahnfabrik, Bad Säckingen, Germany), by placing the tip perpendicular to the sample surfaces with a grey background and natural daylight [26]." [lines 114-121]. Please offer an explanation.
Response: Dear Reviewer, thank you for your question. Our aim was to create a protocol through procedures that had been already published and therefore validated in the literature. This is the reason why we wanted to underline how the following steps had been previously validated: the use of Mylar sheet, the storage timing and temperature, the tip perpendicular to the specimen, and the grey background used. Following your recommendation, we removed most of these references and kept only the final one (Line 113).
Comment 5: Same question for References 30 and 31[lines 139-141]. Do they represent a standard point necessary to be used in the Materials and methods?
Response: Thank you for this observation. We removed references 30 and 31 from lines 132-133.
Comment 6: please check row 277 => there is Reference 41 "[14,27,29,39,41]" and then row 292 -> there is Reference 40 "color stability [40]"
Response: Thank you for your observation. We updated the reference order.
Comment 7: I would suggest revising and rephrasing the following statement: "One of the few articles in which the pigmentation effect of energy drinks has been evaluated on composites was published in 2024" [lines 324-325].
A simple search on Pubmed with the following search words: "energy drink AND colour AND composites resin" will show the information from 2012 at least (Erdemir U, Yildiz E, Eren MM. Effects of sports drinks on color stability of nanofilled and microhybrid composites after long-term immersion. J Dent. 2012 Dec;40 Suppl 2:e55-63. doi: 10.1016/j.jdent.2012.06.002. Epub 2012 Jun 17. PMID: 22713737). See also other related articles in the Pubmed search list.
Response: Thank you for your comment. We modified the entire sentence (Line 268).
Comment 8: References. Please check Ref 40 (row 466)
Response: Done.
Comment 9: please check all the References' styles to be according to the Journal's recommendations
Response: Thank you for the suggestion. We checked all references following the Journal guidelines.
Reviewer 4 Report
Comments and Suggestions for Authors
Dear authors,
congratulations for the chosen topic, of great impact in clinical practice.
-How was the number of specimens determined to validate the results?
-can the results be systematized in tables including p-value?
-from the point of view of recommendations for the patient, do you consider it opportune to formulate them in order to preserve the restoration color?
Author Response
Comment 1: Dear authors, congratulations for the chosen topic, of great impact in clinical practice.
Response: Thank you for your opinion and for all the useful indications.
Comment 2: How was the number of specimens determined to validate the results?
Response: Thank you for this question. At the end of 2022, a study group formed by members of European Federation of Conservative Dentistry and members of the Dental Materials Group of the International Association for Dental Research developed a a risk of bias tool for pre-clinical dental materials research studies (doi: 10.1016/j.jdent.2022.104350). According to this RoBDEMAT risk of bias tool for pre-clinical dental research, one of the accepted ways to determine the sample size is to calculate it depending on a previous study. Therefore, we chose a previously published study to determine our specimens sample size.
We added this explanation (Lines: 140-142).
Comment 3: can the results be systematized in tables including p-value?
Response: Thank you for your question. We acknowledge that Tables 3 and 4 are difficult to follow. To improve clarity: we updated the tables by integrating the presentation of statistically significant differences directly into the table format. This will involve clearly marking significant differences within the table column. These adjustments aim to make the data and statistical differences more comprehensible and easier to interpret.
Comment 4: from the point of view of recommendations for the patient, do you consider it opportune to formulate them in order to preserve the restoration color?
Response: Thank you for your question. We added clinical recommendations in the Conclusions section (Lines 357-361).
Round 2
Reviewer 2 Report
Comments and Suggestions for Authors
All my questions were responded.
Congratulations!
Comments on the Quality of English LanguageJust a revision for publication, nothing more.
Reviewer 4 Report
Comments and Suggestions for Authors
Dear authors,
thank you for the modifications; I believe that they respond to the requests.